# The Molecular Architecture of *Pseudomonas aeruginosa* Quorum-Sensing Inhibitors

**DOI:** 10.3390/md20080488

**Published:** 2022-07-28

**Authors:** Qiaoqiang Li, Shen Mao, Hong Wang, Xinyi Ye

**Affiliations:** Collaborative Innovation Center of Yangtze River Delta Region Green Pharmaceuticals, Key Laboratory of Marine Fishery Resources Exploitment & Utilization of Zhejiang Province, College of Pharmaceutical Science, Zhejiang University of Technology, 18 Chaowang Road, Hangzhou 310014, China; 2112023022@zjut.edu.cn (Q.L.); 1111923009@zjut.edu.cn (S.M.)

**Keywords:** quorum sensing, *Pseudomonas aeruginosa*, inhibitor, virulence

## Abstract

The survival selection pressure caused by antibiotic-mediated bactericidal and bacteriostatic activity is one of the important inducements for bacteria to develop drug resistance. Bacteria gain drug resistance through spontaneous mutation so as to achieve the goals of survival and reproduction. Quorum sensing (QS) is an intercellular communication system based on cell density that can regulate bacterial virulence and biofilm formation. The secretion of more than 30 virulence factors of *P. aeruginosa* is controlled by QS, and the formation and diffusion of biofilm is an important mechanism causing the multidrug resistance of *P. aeruginosa*, which is also closely related to the QS system. There are three main QS systems in *P. aeruginosa*: *las* system, *rhl* system, and *pqs* system. Quorum-sensing inhibitors (QSIs) can reduce the toxicity of bacteria without affecting the growth and enhance the sensitivity of bacterial biofilms to antibiotic treatment. These characteristics make QSIs a popular topic for research and development in the field of anti-infection. This paper reviews the research progress of the *P. aeruginosa* quorum-sensing system and QSIs, targeting three QS systems, which will provide help for the future research and development of novel quorum-sensing inhibitors.

## 1. Introduction

To battle against infection, antibiotics are often used to kill pathogenic bacteria. However, with the abuse and misuse of antibiotics, bacteria increasingly exhibit resistance to available antimicrobial drugs [1,2]. Hence, it is urgent to find novel strategies to tackle the new generation of drug-resistant pathogens. Quorum sensing (QS) is a well-known cell-to-cell signal communication system that allows bacteria to monitor their cell density by releasing signaling molecules called autoinducers (AIs) to cope with the changes in society and environment [3,4]. QS conducts and regulates the expression of related genes through signal molecules, such as bioluminescence, pigment production, formation of biofilm, and the secretion of virulence factors [5].

As the core of the QS system, signal molecules play a key role in the signal transmission process. It has been reported that blocking the QS system of bacteria can significantly reduce the pathogenicity of pathogens without affecting their growth. Therefore, targeting the QS system to treat bacterial infections provides a new direction for effectively slowing down the development of bacterial resistance. The initial demonstration of quorum-sensing inhibitors (QSIs) can be traced back to Australian red marine algae *Delisea pulchra*. It produces secondary metabolites, a number of halogenated furanones that have structural similarities to AHL molecules. These metabolites can interfere with the bacterial processes that involve AHL-driven quorum-sensing systems [6]. To date, many compounds from various natural sources, molecules with structural similarities to quorum-sensing signals and synthetic chemicals have been reported to act as quorum-sensing inhibitors. In recent years, more and more researchers are devoted to the study of QS, which is meaningful for developing novel inhibitors.

*Pseudomonas aeruginosa* (*P. aeruginosa* or PA) is a common opportunistic Gram-negative bacterium in the clinic [7] that can cause various kinds of infections in immune-compromised patients, including individuals with cancer, cystic fibrosis, HIV and burn victims, etc. [8]. In *P. aeruginosa*, the production of virulence factors, motility, and bioluminescence as well as biofilm formation, are mainly controlled by QS [9]. Hence, blocking QS system is expected to be a new way to treat *Pseudomonas aeruginosa* infection.

There are three main QS systems in *P. aeruginosa*, two of which use *N*-acylhomoserine lactones (AHLs) as signal molecules, called the *las* system and *rhl* system, respectively [10]. The third QS system is termed the *p**qs* system, and it is interlinked with the other two systems. All the quorum-sensing systems are closely related to each other and do not exist independently. These three systems together form an intricate hierarchical quorum-sensing circuitry (Figure 1) [11]. Among the three QS systems, *las* is at the top of the QS hierarchy and is required for the optimal activation of the *rhl* and *pqs* QS systems [12].

There are many reports on the inhibitors [13,14,15,16,17,18] of *P. aeruginosa*; flavonoids, allicin analogs from plants, equisetin isolated from marine fungus, and so on can show potent inhibitory activity. Existing drugs also deserve further study; for example, studies showed that piroxicam had potential QS inhibitory activity. Thus, reusing old drugs is an important method of developing new QSIs. Regarding the working mechanisms, some of them have been studied clearly, but there are still many compounds that can act as inhibitors with unclear action mechanisms. This paper reviews the research progress of quorum-sensing inhibitors against *P. aeruginosa* from the aspects of screening from various sources and working mechanisms.

## 2. Inhibitors of the *las* System

### 2.1. The Introduction of las 

*Las* plays a fundamental role in the regulation of the QS network of *P. aeruginosa*; the signaling molecule *N*-(3-oxo-dodecanoyl)-L-homoserine lactone (3-oxo-C12-HSL, OdDHL) is synthesized by LasI. When the signal concentration reaches the threshold, it binds to the receptor protein LasR to form a complex and initiates the expression of downstream target genes [19]. The *las* signaling system regulates virulence factors such as exotoxin A and the expression of LasA staphylolytic protease, LasB elastase, and Apr alkaline protease as well as biofilm formation [20]. LasA and LasB play a vital role in regulating protease activity; they are in charge of the degradation of elastin, fibrin, complement, interferon, and collagen during infection. They can also influence toxin levels [21]. Therefore, suppressing *las* is an effective way to inhibit the QS system. There have been many reports about *las* inhibitors including molecules from natural plant extracts, similar structures of AHL signals, and artificial compounds. *Las* pathways involved with the QS controls of virulence and other phenotypes are shown in Figure 2.

### 2.2. Inhibitors Acting on las 

Currently, inhibitors of the *las* system are the most studied. Specific *las* inhibition is measured based on expression levels of the respective genes through various approaches, such as RNA extraction and microarray analysis, high-throughput RNA sequencing, molecular docking, gene expression assay, and quantitative real-time PCR. Specific *rhl* and *pqs* inhibition are measured through similar approaches to those for *las*. Many researchers found that some natural materials from different sources had QS inhibitory activities by targeting *las* and most are from plants. The results showed that ethyl acetate extract of TemuIreng rhizomes (*Curcuma aeruginosa*) can decrease *P. aeruginosa* virulence responses such as protease LasA, LasB, and biofilm formation, which is controlled by quorum sensing [21]. As we all know, many organosulfur compounds have bio-inhibitory activities, so many researchers have devoted effort to studying the related compounds. It was reported that garlic-derived natural vinyl dithiins **1** and **2** (Figure 3a) inhibited LasI/LasR-based QS but exclusively in LuxR monitoring system [13]. Sulforaphane and erucin (Figure 3a) are natural isothiocyanates from broccoli. They can effectively combine with LasR to inhibit the QS system [22]. Many essential oils also can serve as inhibitors such as the oil of *Murraya koenigii*, of which the components can target *las* and thereby inhibit the production of pigment and siderophore in *P. aeruginosa* [23]. Bettadaiah et al. [14] found that the key phenolic compounds of ginger; 6-gingerol; 6-shogaol; zingerone; a derivative of 6-shogaol, 6-azashogaol; and an isoxazoline derivative of 6-gingerol (Figure 3b) showed good QSI activities against *P. aeruginosa*. These compounds, consisting of long alkyl chains and moieties of *N*-acyl-homoserine lactone, can inhibit LasR-dependent gene expression. Among these compounds, 6-gingerol showed the best QSI activity against *P. aeruginosa*. Many flavonoids were also proved to have QSI activities and can be inhibitors. Quercetin (Figure 3c) is a flavonoid that interacts with transcriptional regulator LasR as well as suppressing QS circuitry, biofilm formation, and virulence gene expression [15]. The structure of naringenin is similar to quercetin, so it plays a similar role to the latter. It showed its effect by directly binding the QS regulator LasR and competing with 3-oxo-C12-HSL. The compound reduced the production of AHLs in *P. aeruginosa* and therefore inhibited the production of QS-regulated virulence factors [16]. Studies have shown that other flavonoids (Figure 3c) such as catechin-7-xyloside (C7X, a flavan-3-ols from *Spiraea hypericifolia*), sappanol (a 3,4-dihydroxyhomoisoflavan from *Caesalpinia sappan*), and butein (a chalcone from *Toxicodendron vernicifluum*) were capable of interacting with LasR of *P. aeruginosa* (a LuxR-type quorum-sensing regulator) without killing the bacteria [24]. *Phyllanthus emblica* is a traditional medicinal plant that has various biological activities. Docking studies showed that its component cardamonin belonged to the chalcone compounds and might have good inhibitory activity against QS by binding with LasR receptor [25].

Other natural products have also been reported to have inhibitory activities (Figure 4). Furvina is a nitro-vinylfuran antibiotic that was first isolated from bagasse in Cuba. It was reported that it can interact with *las* as well as interfere with 3-oxo-C12-HSL signal molecule and bioluminescence emission [26]. Therefore, it can be a promising complementary strategy that furvina might be applied to combine with less effective antibiotics to cure biofilm-related infections. *Trans*-anethole is the major component of anise (*Pimpinella anisum*) oil, having similar structure to that of natural ligand OdDHL, and it shows antimicrobial and antifungal activities [27]. Moreover, it was demonstrated that *trans*-anethole reduced the production of virulence factor by binding to LasR protein [27]. 1,8-Cineole is the main component of niaouli essential oil, and it shows an inhibitory effect on biofilm formation in *P. aeruginosa*. A molecular docking study confirmed that it interfered the binding with LasR and inhibited *las* [28]. Some catabolites were reported to inhibit quorum sensing by inhibiting *las*. For example, 4-methylenebut-2-en-4-olide, a protoanemonin, is produced by the pseudomonads *Pseudomonas* sp. B13 and *Pseudomonas reinekei* MT1. It inhibits a *lasB*-based QS monitor and the production of elastase LasB and phenazine [29]. Because many natural products can be used as QSIs, researchers tried to synthesize new compounds by changing the structure of natural products. (*R*)-bgugaine has antibacterial activity, while synthesized norbgugaine is a demethylated form of natural bgugaine (*R*)-norbgugaine showed inhibition to various motilities, pyocyanin pigmentation, the production of LasA protease, and biofilm formation in *P. aeruginosa* [30]. (4-((E)-(4-hydroxy-2-methylphenylimino)methyl)-2-methoxyphenol (MMP) is a vanillin derivative. After in-depth analysis, it was found to have a strong binding affinity to the *las* system [31].

To develop more QSI against *P. aeruginosa*, some researchers attempted to design inhibitors based on the AHL signaling molecules. The structures of *N*-(4-{4-fluoroanilno}butanoyl)-L-homoserine lactone (FABHL) and *N*-(4-{4-chlororoanilno}butanoyl)-l-homoserine lactone (CABHL) are similar to HSLs (Figure 5a). These two compounds can influence *las* via interacting with LasR protein [32]. Based on the fact that chalcones have widespread biological activity, Wu et al. focused on central amide moieties and synthesized a series of chalcone-based l-homoserine lactones; among these compounds, (*S*)-2-((4-(3-(4-bromo-2-fluorophenyl)acryloyl)phenyl)amino)-*N*-(2-oxotetrahydrofur-an-3-yl) acetamide **3** (Figure 5a) was found to inhibit the LasR-dependent QS system of *P. aeruginosa* [33]. Over the past years, designing new QSIs by modifying the structure of signaling molecules has always been a widely used approach. In 2011, *N*^1^-(5-chloro-2-hydroxyphenyl)-*N*^3^-octylmalonamide **4** (Figure 5b) was developed structurally in the L-HSL ring system. It was modified by replacing the methylene group to β-ketoamide in the acyl chain of OdDHL with an NH group, and at the same time, the replacement of the homoserine lactone system with stable cyclic groups can be conducted [34]. This compound displayed LasR-induction antagonist activity and influenced the host immune system. In 2013, an OdDHL analogue **5** (Figure 5b) containing a nonnative head group was found to be a potent inhibitor of LasR-based quorum sensing, reducing the production of pyocyanin by disrupting LasR [35]. In 2015, Byun et al. [36] designed several novel QS compounds by replacing the lactone ring of OdDHL with the pyrone ring. By varying alkyl chain lengths to inhibit the binding between OdDHL and LasR of *P. aeruginosa*, **6** (Figure 5b) was the most potent compound that displayed strong inhibitory activities against biofilm formation. In the same year, **7** (V-06-018, Figure 5b) was designed to modulate LasR strongly and influence QS-dependent phenotypes in *P. aeruginosa* [37]. In 2017, it was investigated that (*S,E*)-2-hydroxy-*N*-(2-hydroxy-5-nitrobenzylidene)propanehydrazide(lactohydrazone) **8** (Figure 5b) displayed interactions as a natural auto-inducer with LasR and then inhibited virulence factors [38]. The indole-based AHL analogue **9** (Figure 5c) and salicylic acid (SA) (Figure 5c) can effectively inhibit the *P. aeruginosa* LasR. Accordingly, in 2018, Sekhar et al. designed and synthesized some derivatives of 2-phenylindole-amide-triazole and salicylic acid-triazole; among the compounds, four (**1****0a**–**c**, **1****1**) were found to have inhibitory activity (Figure 5c) [39]. Later, the same group developed **1****2a**–**d** (Figure 5c) as novel inhibitors of LasR-dependent quorum sensing; these compounds exhibited quorum-sensing inhibitory activity without showing toxic effects on normal cells [40].

Some other researchers tried to focus on some different scaffolds and functional groups. Lactams were designed and synthesized through the reductive amination of mucochloric acid and mucobromic acid. **1****3a**–**d** (Figure 6a) were found to be the most active compounds and can dock to the LasR receptor protein and serve as new lead compounds for developing potent QSI compounds [41]. Triphenyl **1****4a** and a nonnatural triaryl analogue **1****4b** (Figure 6a) containing chlorine atom show high LasR antagonist activity [42]. Nonnatural irreversible antagonists **1****5a** and **1****5b** (Figure 6a) were identified to exhibit effective LasR antagonist activity and inhibit the expression of virulence factors such as pyocyanin and the formation of biofilm [43]. Based on triphenyl scaffold, some researchers designed various new inhibitors to strongly agonize LasR, and then they found that **1****6** (Figure 6a) was the most potent LasR antagonist [44]. Computer-aided approaches were also a good way to develop novel compounds; some researchers designed five new compounds (**1****7a**–**e**) (Figure 6b); these can act as inhibitors of the LasR QS system of *P. aeruginosa* in a dose-dependent manner [45]. Novel compounds containing benzothiazole moiety were synthesized. Among them, **1****8a**–**c** (Figure 6b) have more potent inhibitory activity against LasR in *P. aeruginosa* [46]. A series of unsymmetrical azines were designed and evaluated, and docking analysis showed that **19a** and **19b** (Figure 6b) can bind to LasR protein [10]. Celecoxib derivatives **SGK 324**, **327** and **330** (Figure 6c) have favorable interactions with LasR binding sites. Among these compounds, **SGK 330** is the most active because at the place of the lactone ring of the natural ligand HSL, the interaction site was occupied by the thiazolidinone ring [47]. A range of dihydropyrrolone analogues were synthesized and can dock to LasR, and **2****0** (Figure 6c) was the most potent inhibitor of biofilm formation [48]. Two novel Mannich bases, 1-(phenyl (o-tolylamino) methyl) urea **2****1** and 3-((1H-benzo[*d*]imidazol-1-yl)methyl)naphthalen-2-ol **2****2** (Figure 6c), were synthesized by Mannich reaction, showing LasR antagonistic activities and inhibiting biofilm formation and pyocyanin production in a dose-dependent manner [49]. Chrysin derivative **2****3** (Figure 6c) can bind to regulator LasR and shows anti-virulence activity [50]. Employing a novel specialized multilevel in silico approach, plenty of compounds were assessed, and eight compounds (**2****4****a**–**h**) (Figure 6d) were identified as promising LasR inhibitors [51].

## 3. Inhibitors of the *rhl* System

### 3.1. The Introduction of rhl 

Another system belonging to the LuxR-type receptor family in the QS system of *P. aeruginosa* is *rhl*, which is similar to *las*. The difference is that the AHL-signaling molecule of *rhl* is *N*-butanoyl-L-homoserine lactone (BHL or C4-HSL), which is produced by RhlI, and the signal molecule binds to its receptor and activates transcriptional regulator RhlR [52]. *Rhl* is related to the production of some virulence factors such as rhamnolipids and the toxic exo-factors hydrogen cyanide, pyocyanin, and so on [53]. Among these, rhamnolipids are biodegradable surfactants that contain two fatty acid molecules and rhamnose residues [54]. They play a vital role in protecting cells from oxidative stress and participating in biofilm growth and maturation [55]. Pyocyanin, a greenish pigment secreted by *P. aeruginosa*, can cause severe toxic effects on the host [56]. Therefore, targeting *rhl* is a good strategy for controlling quorum sensing infections. *Rhl* pathways involved with the QS controls of virulence and other phenotypes are shown in Figure 7.

### 3.2. Inhibitors Acting on rhl 

Inhibitors acting on *rhl* systems can be divided into three sources: natural products, analogues of signaling molecules, and synthetic compounds. Some inhibitors are from natural products. Three ellagic acid derivatives (**2****5****a**–**c**, Figure 8a) isolated from *Anogeissus leiocarpus* (DC) have been reported to inhibit the expression of QS-regulated virulence factors and down-regulate the QS regulator gene *rhlR* [1]. Tannic acid (TA) (Figure 8a) was tested to suppress the synthesis of pyocyanin, which is controlled by *rhl*, including the transcriptional activator RhlR and the auto-inducer synthase RhlI [57]. The plant compound rosmarinic acid (RA) (Figure 8a) was shown to specifically bind to the RhlR QS receptor of *P. aeruginosa* and induce some QS-mediated behaviors [58]. Some are analogues of signaling molecules. It was identified that mBTL (Figure 8b), an analogue of the native *P. aeruginosa* autoinducers, can inhibit the quorum-sensing receptor RhlR and prevent biofilm formation [59]. A series of hybrid AHL analogs was designed, and among those evaluated compounds, *N*-acyl-L-homocysteine thiolactones **2****6a**–**c** (Figure 8b), which contained homocysteine thiolactone head groups, were identified to be promising compounds that showed high selectivity for RhlR [60]. On the other side, synthetic compounds such as 2, 8-bit derivatives of quinoline synthesized by Suzuki coupling reaction were found to be potent inhibitors against biofilm formation. After optimization by Sun and co-workers, **2****7a** and **2****7b** (Figure 8c) showed the most potential candidates mainly inhibiting the expression of *rhl* [61]. A set of 4-gingerol analogues were synthesized, and structure-activity relationship studies identified that the alkynyl ketone **2****8** (Figure 8c) can act as a strong RhlR antagonist and shows inhibitory activities on biofilm formation and rhamnolipid production in *P. aeruginosa* [62].

## 4. Inhibitors of the *pqs* System

### 4.1. The Introduction of pqs 

The third QS system in *P. aeruginosa* is called *pqs*, and it is closely related to two AHL-based systems. It puts 2-heptyl-3-hydroxy-4-quinolone (PQS) as its signal molecules, and the precursor of PQS, 2-heptyl-4-hydroxyquinoline (HHQ), is commonly associated with QS along with *pqs* [63]. The biosynthesis of these two signal molecules requires a transcriptional regulator PqsR (MvfR) and *pqsABCDE* operon, whose expression is controlled by the former [64]. PqsR plays a key role in *pqs* and regulates the expression of *rhlI* as well as some virulence functions such as hydrogen cyanide, pyocyanin, bacterial motility, and biofilm formation, of which the expression is activated by LasR but inhibited by RhlR [65]. Hence, PqsR is a potential target for the study of novel QS inhibitors. *Pqs* pathways involved with the QS controls of virulence and other phenotypes are shown in Figure 9.

### 4.2. Inhibitors Acting on pqs

In recent years, increasing numbers of researchers have paid attention to QS inhibitors targeting *pqs*. There are some inhibitors from natural sources such as stigmatellin Y and wogonin (Figure 10), with similar structural skeletons. Stigmatellin Y was from *Bacillus subtilis* BR4, and it showed strong affinity to PqsR as well as competing with *pqs* to distract PQS-PqsR mediated communication [66]. Wogonin is the active ingredient from *Agrimonia pilosa*. One study showed that it can target the PqsR of *pqs* and be a potent QS inhibitor [67]. Farnesol (Figure 10) is a natural sesquiterpene alcohol, and it can suppress the transcription and protein expression of the *pqsABCDE* and *pqsH* genes and interfere with the synthesis of pyocyanin [68]. Perillaldehyde (Figure 10) is a natural, widely used, and nontoxic food additive from *Perilla frutescens*. It showed anti-virulence activity and repressed biofilm formation. Molecular mechanism studies showed that it can interact with PqsR protein [69].

However, many QS inhibitors still rely on synthesis. Hartmann et al. have reported that antagonists of PqsR (**29a** and **29b**, Figure 11a) can inhibit the production of pyocyanin and may have potential to become QS inhibitors [70]. The thiazolyl-containing quinazolinones **3****0a** and **3****0b** (Figure 11a) play similar roles as **29a** and **29b**, and they were demonstrated to attenuate the production of pyocyanin and to reduce many other virulence responses via targeting PqsR [71]. Quinazolinone **3****1** (Figure 11a) can also act as a PqsR inhibitor of biofilm formation in *P. aeruginosa* [72]. Two series of chromones-based compounds were synthesized with the help of molecular docking. Based on the bioactive results, **3****2** (Figure 11a) was the most potent inhibitor of biofilm; it showed good predicted affinities for PqsR and may be a promising QS inhibitor targeting *pqs* [73]. Many compounds containing amide bonds or ureido motifs can act as inhibitors. It was identified that ureidothiophene-2-carboxylic acids **3****3** (Figure 11b) can be inhibitors of PqsD. Through combining in silico analysis and biophysical methods, **3****4** was synthesized based on **3****3** (Figure 11b), which carried a phenylalanine substituent at the ureido motif as the most active inhibitor [64,74]. 3-Cl substituted benzamidobenzoic acid **3****5** (Figure 11b) can also be used as an inhibitor of PqsD, playing a key role in the QS system and mediating the formation of HHQ [75]. Through a scaffold-based approach, Hartmann et al. developed a novel inhibitor of PqsD **3****6** (Figure 11c). It reduced the HHQ and PQS levels and significantly inhibited the formation of biofilm with no antibiotic effects [76]. PqsD shared some features with chalcone synthase, and thereby, a novel class of inhibitors composed of catechol moieties was discovered. Among those compounds, **3****7a**, **3****7b**, and **3****7c** (Figure 11c) showed inhibitory activities [77]. A series of N-1, C-2, C-6-substituted 3-hydroxy-pyridin-4(1H)-one derivatives were synthesized and evaluated, **3****8** (Figure 11c) was found to be the most active compound, it strongly suppressed biofilm formation at low concentrations through inhibiting the expression of PqsA and reducing the production of virulence factors. It was considered a potentially novel QS inhibitor [78]. Compounds containing aminoquinolines were reported to have inhibitory activities. Quinolone-derived molecule **39** (Figure 11d) was identified to show inhibitory activity in the production of alkyl quinolone (AQ), a *pqs* signal molecule [79]. It was found that the 7-Cl and 7-CF_3_ substituted long-chain *N*-dodecylamino-4-aminoquinolines **4****0a** and **4****0b** (Figure 11d) can act as PQS antagonists [3]. A series of *N*-octaneamino-4-amino-quinoline derivatives were also synthesized, such as **4****1a** and **4****1b** (Figure 11d) with benzofuran and benzothiophene substituents respectively; they showed the effective inhibition of the production of pyocyanin and can reduce the motility of *P. aeruginosa* via interfering with the *pqs* signaling pathway [65]. **M64** (Figure 11d), one of the benzamide–benzimidazole (BB) series compounds, was the first PqsR inhibitor to show in vivo activity; it targeted PqsR and interfered with the early steps of biofilm formation [80]. Through further optimization, **4****2** (Figure 11d) was designed and reported to act as a potent PqsR antagonist, inhibiting the production of HHQ and PQS at sub-micromolar concentrations [81]. **4****3** (Figure 11e) is a novel lead QSI and shows high potency, targeting the PqsR of *pqs* and inhibits the production of pyocyanin [82]. Recently a study showed that vitamin E and vitamin K_1_ (Figure 11e) significantly inhibited the formation of biofilm and the production of virulence factors, which had potential as QS inhibitors through inhibiting PqsR protein [83].

In recent years, many antibacterial drugs have been developed, some of which can be used as QSIs. The FDA-approved antivirulence drugs nitrofurazone and erythromycin estolate (Figure 12), which belong to the nitrofuran and macrolide structural classes, respectively, are able to inhibit the main effector protein PqsE of *pqs* and reduce the production of the PqsE-controlled virulence factors such as pyocyanin and rhamnolipids [17]. Some researchers have applied a combined multilevel computational approach to find possible QS inhibitors that target the PqsR of *P. aeruginosa* in FDA-approved drugs. They found that nilotinib, indocyanine green, cabozantinib, venetoclax, and montelukast (Figure 12) can act as QS inhibitors [84]. In short, present drugs may have new uses and are promising candidates for QS inhibitors.

## 5. Inhibitors Acting on Multiple QS Systems

*Las* is closely connected with *rhl* and *pqs*. It is known that many inhibitors can show inhibition of both *las* and *rhl*. Some inhibitors belong to natural products or their derivatives. The human sex hormones hordenine and estrone (Figure 13a) as well as their structural relatives estriol and estradiol (Figure 13a) were found to decrease AHL accumulation and the expression of six QS-regulated genes (*lasI*, *lasR*, *lasB*, *rhlI*, *rhlR*, and *rhlA*) in *P. aeruginosa* [85]. Researchers have reported that many natural flavonoids had inhibitory activities. For instance, 3,5,7-trihydroxyflavone (TF, Figure 13b) from *Alstoniascholaris* leaf extract and luteolin from traditional Chinese herbal medicine were found to inhibit the QS signaling molecules OdDHL in *las* and BHL in *rhl* through molecular docking study; they had inhibitory effect on the formation of biofilm and the production of QS-dependent virulence factors [86,87]. Baicalein (BCL) and naringin (Figure 13b) also belong to natural flavonoids and play similar roles to those of TF and luteolin in *P. aeruginosa* [8,88]. 6-methylcoumarin (6-MC, Figure 13b) is one of the coumarin derivatives. Molecular docking studies showed that it inhibited LasI in *las* and RhlI/RhlR in *rhl* [89]. Natural products of other structures also exhibit some inhibitory abilities. The isoquinoline alkaloid berberine and polyphenol chlorogenic acid (Figure 13c) were extracted from plants, and they displayed significant inhibition of quorum-sensing-regulated phenotypes. Molecular docking analysis showed that these two compounds appeared to be potent inhibitors of LasR and RhlR [8,90]. The natural pentacyclic triterpenes betulin (BT, Figure 13c) and betulinic acid (BA, Figure 13c) can exhibit significant attenuation of the QS receptors, LasR, and RhlR [88]. A triterpenoid coumarate ester isolated from *Dalbergia trichocarpa* has been identified as oleanolic aldehyde coumarate (OALC, Figure 13c). Evernic acid (Figure 13c) was a lichen secondary metabolite that can be isolated from *Evernia* species. These two compounds can inhibit the expression of QS-regulated *lasB* genes and *rhlA* genes and reduce virulence factors [91,92]. 1-(4-amino-2-hydroxyphenyl)ethanone (AHE, Figure 13c) was identified from the crude extract of *Phomopsis liquidambari*S47, and it had strong binding affinity with LasR and RhlR and repressed the transcriptional levels of LasI and RhlI [93]. Many natural plant extracts, for example, the extract of *Dalbergia trichocarpa* bark (DTB) and aqueous extract of Yunnan Baiyao, showed significant interference with *las* and *rhl* of QS signaling circuit [94,95]. There are four phytoconstituents from *Cassia fistula* influence AHL activity, rhein, 3-aminodibenzofuran, 5-(hydroxymethyl)-2-(dimethoxymethyl)furan and dihydrorhodamine (Figure 13d). They target *las* and *rhl*, down-regulate the two system-related genes, and have effects on biofilm formation [96].

Gingerol analogues **4****4a** and **4****4b** (Figure 14a) can reduce biofilm formation effectively and have potent binding affinity to LasR by increasing acyl chain length and modifying absolute configuration [97]. Meanwhile, it can bind to another QS receptor RhlR. Potassium 2-methoxy-4-vinylphenolate **4****5** (Figure 14a) has a partially similar structure to that of 6-gingerol, and it was found to target *P. aeruginosa*
*las* and *rhl* circuitry and to inhibit biofilm formation along with various virulence factors such as LasA protease, LasB elastase and pyocyanin [98]. The potential active compound **4****6** (Figure 14a) isolated from the extract of *Delftia tsuruhatensis* SJ01 was identified as 1,2-benzenedicarboxylic acid diisooctylester, having a similar structure with AHL. It competes with LasR and then down-regulates the protease and elastase activity along with the *rhl* system expression [99]. In 2012, a novel QS inhibitor C2 (*N*-decanoyl-L-homoserine benzyl ester) (Figure 14b), which belongs to the structural analogues of AHLs, was found to repress the expression of *lasR*, *lasI*, *rhlR*, and *rhlI* to varying degrees and to inhibit *las* and *rhl* [100]. Some compounds containing five-membered rings also show inhibitory activities. Terrein was first isolated from *Aspergillus terreus* as a secondary bioactive metabolite, 5-hydroxymethylfurfural (5-HMF) (Figure 14b); it can be found in honey, dried fruits, wine, coffee, and so on. These two compounds antagonize both regulatory proteins LasR and RhlR, inhibiting the production of virulence factors and the formation of biofilm [101,102]. Moreover, the extracts of *Quercus infectoria* gall, *Hypericum perforatum* L., and olive leaf were also revealed to reduce the expression of *las* and *rhl* related genes levels [56,103]. Other extracts, for instance *Senna alexandriana* mill extracted by acetone and hexane; 3-isobutylhexahydropyrrolo[1,2-a] pyrazine-1,4-dione (**4****7**, Figure 14b), a component of *Phomopsis tersa*, a kind of cranberry extract rich in proanthocyanidins (cerPAC); and chitosan extracted from *Aspergillus flavus* were demonstrated to interfere with both the *rhl* pathway and the *las* pathway [19,104,105]. Several reported drugs exhibit QS inhibitory activities. Ibuprofen belongs to the NSAID compounds, and it is one of the most popular nonprescription drugs; it was discovered that ibuprofen can bind with LuxR, LasR, LasI, and RhlR at high binding scores and inhibit both *las* and *rhl* as well as biofilm formation in *P. aeruginosa* [106]. Aspirin can cause significant reductions of elastase, protease, and so on at sub-MICs, and the antagonist activity of aspirin can be attributed to *las* and *rhl* [107]. More recently, there was a new discovery that the FDA-approved drug allopurinol can act as a QS inhibitor against *P. aeruginosa*, and through molecular docking study, it exerted inhibitory effects by binding LasR and RhlR receptors [108].

Amorfrutins B and synthesized **4****8a**–**c** (Figure 15a) were analogues of the natural products cajaninstilbene acid (CSA). They can inhibit the *las* system and *las* system-related virulence factors by competitively binding with the receptor OdDHL; they can also interrupt *pqs* and show biofilm inhibitory activity [109,110]. It was found that a caffeic acid derivative with phenolic hydroxyl group **49** (Figure 15a) played the same role as amorfrutins B. It inhibits *las* and *pqs* as well and has favorable inhibition of the expression of virulence factors [111]. Through the research and exploration of Chinese herbal medicine, falcarindiol (Figure 15b), the main ingredient of *Notopterygiumincisum*, was found to have inhibitory effects on both *las* and *pqs*. Molecular docking analysis revealed that it competed with natural ligands and combined with LasR [112]. Synthetic furanone C-56 and furanone C-30 (Figure 15b) are analogues of AHL signaling molecule. They were reported to interfere with *N*-acylhomoserine lactone and had potent inhibitory ability against QS in *P. aeruginosa* [113]. According to the previous reports, dihydropyrrol-2-ones (DHP) molecules can be potential QS inhibitors, so both halogenated **5****0a**–**c** and non-halogenated **5****0d** (Figure 15b) compounds were evaluated. They showed good inhibition of LasR and PqsR receptors; hence, it inhibited *las* and *pqs* [114]. Cefoperazone (CFP, Figure 15b) is one of the most widely used β-lactam antibiotics. Some researchers tried to explore some other activities of CFP, and its metallic derivatives cefoperazone iron complex (CFPF) and cefoperazone cobalt complex (CFPC) were found to interact more strongly with LasI, LasR, and PqsR receptors than CFP, CFPF, and CFPC and to eliminated QS-associated virulence factors [115]. Hadizadeh et al. found that oxicams (piroxicam and meloxicam, Figure 15b) can serve as the best inhibitors of LasR from the studied nonsteroidal anti-inflammatory drug (NSAID) compounds and also have potential antagonist activity against PqsE in the *pqs* system [116].

There were also some other compounds that inhibited both *rhl* and *pqs*. 3-phenyllactic acid (PLA**,**
Figure 15c) is produced by *Lactobacillus* species and strongly binds to receptors RhlR and PqsR. **S4** (Figure 15c) was designed to disrupt *rhl*–*pqs* crosstalk [117,118]. In addition, sulfur-containing compounds tend to have inhibitory activity. Allicin (diallyl thiosulfinate) (Figure 15c), which was extracted from the edible plant garlic, was found to inhibit C4-HSLs synthesis-related gene *rhlI*. It also suppresses the expression levels of genes like *rhlA*, *rhlB*, and *rhlC*, which is related to rhamnolipid synthesis and inhibits PQS molecule synthesis [119].

In many cases, the three systems have mutual influence, and partial inhibitors can work on the three systems at the same time. Some are from natural sources. Baicalin (an active natural flavonoid extracted from the traditional Chinese medicinal *Scutellaria baicalensis*, Figure 16a), soy isoflavones (a group of phenolic compounds rich in soybeans and other legumes, Figure 16a), epigallocatechin-3-gallate (EGCG, a bioactive component of green tea, Figure 16a), and 1H-pyrrole-2-carboxylic acid **5****1** (a compound isolated from *Streptomyces coelicoflavus*, Figure 16a) showed inhibitory activities to *las*, *rhl*, and *pqs* and attenuated various virulence phenotypes (Figure 16a) [120,121,122,123]. The bacterial extract rhodamine isothiocyanate (Figure 16a) is a potential bioactive compound that can downregulate QS regulatory genes and virulence-related genes [124]. As we all know, ajoene from garlic can inhibit quorum sensing. On the basis of mimicking the active skeleton of ajoene, Givskov et al. synthesized a series of sulfur-containing ajoene-derived compounds by simply changing substituents. Among those compounds, **5****2a** and **5****2b** (Figure 16b) display efficient inhibitory activity to the *las*–*rhl*–*pqs* system without affecting bacterial activity [125]. Diallyl sulfide (DAS) and diallyl disulfide (DADS) (Figure 16b) from garlic oil work in the same way and inhibit the essential gene expression of the three QS systems and the production of virulence factors [126]. Other compounds such as L-homoserine lactone analogues and phenylurea-containing *N*-dithiocarbamated homoserine lactones, especially **5****3** (Figure 16c), can selectively attenuate the expression of virulence factors, swarming motility and biofilm formation. **53** can competed with OdDHL in binding to LasR by molecular docking analysis and also inhibited the expression of the *las* system genes, *rhl* system genes, and *pqs* system genes [127]. The synthesized itaconimides **5****4a** and **5****4b** and EDTA (Figure 16c) were used as antivirulence compounds that suppress the *las*, *rhl*, and *pqs* systems of *P. aeruginosa* [128]. Sodium ascorbate is a vitamin that may act as an AHL analogue; it shows strong inhibition of the expression of LasR in the *las* system, reduces related virulence factors, and then inhibits *rhl* and *pqs* [129]. *D*-cycloserine has a similar parent nucleus structure with the AHL signal molecule, so it effectively suppresses *las*, *rhl*, and *pqs* [130].

## 6. Inhibition Mechanism Undetermined

There are also many other compounds that can inhibit QS, but their working mechanisms are not yet clear. For instance, the flavone compound chrysin (*Parkia javanica* fruit extract) and the anthraquinone derivative emodin (extracted from rhubarb) were found to have significant antibiofilm activities against *P. aeruginosa* (Figure 17a) [7,131]. Studies showed that terpinen-4-ol (Figure 17b), the main constituent of *Melaleuca alternifolia* essential oil, displayed inhibitory activity to QS-mediated virulence factors and biofilm formation [132]. Some phenolic compounds such as carvacrol from the oregano essential oil and active compounds tyramine and *N*-acetyltyramine from marine bacteria (Figure 17b) can be potential quorum-sensing inhibitors by reducing both pyocyanin production and biofilm formation [133]. The plant-sourced unsaturated fatty acid linoleic acid (LA, Figure 17c) was found to inhibit the formation of biofilm without inhibiting the growth of *P. aeruginosa* [134]. Three sesquiterpene lactones (SLs), **5****5a**–**c** (Figure 17c) isolated from Argentine herb *Centratherum punctatum* were reported to show the QS inhibition of *P. aeruginosa* [135]. The main pyranoanthocyanin **5****6a** (carboxypyranoanthocyanin, red wine extract) and the pure compound carboxypyranocyanidin-3-O-glucoside **5****6b** (Figure 17d) significantly interfered with the expression of several QS-related genes in *P. aeruginosa* and have potential as QS inhibitors [136].

Another possibility being investigated is synthetic compounds. Compounds with a benzene ring such as three hydrazine-carboxamide hybrid derivatives **5****7a**–**c** and the synthetic molecule isoeugenol (Figure 18a) show anti-biofilm activity against *P. aeruginosa* [137,138]. Additionally, a benzalacetone analog 4-methoxybenzalacetone shows the suppression of some virulence phenotypes of *P. aeruginosa* [139]. Studies reported that the indole derivatives 7-fluoroindole, 7-hydroxyindole, and 3-indolylacetonitrile (Figure 18a) were potent antivirulence compounds. They reduced the production of QS-regulated virulence factors in *P. aeruginosa* [39]. Benzimidazole derivatives 5,6-dimethyl 2-aminobenzimidazole **5****8** and two benzimidazolium salts **59a**, **59b** (Figure 18b) were found to have inhibition effects on the production of QS-regulated virulence factors and biofilm formation in *P. aeruginosa* [140,141]. Glyoxamide derivatives **6****0a**–**e** and **6****1** (Figure 18c) display great quorum-sensing inhibition activity and exhibit the inhibition of *P. aeruginosa* biofilms [142]. Some compounds containing a benzene ring and a five-membered heterocycle can possess QS inhibitory activities. For example, 4-(o-methoxyphenyl)-2-aminothiazole (MPAT), 3-amino-2-oxazolidinone derivative YXL-13, and nitrofuran antibiotic furazolidone were identified as showing potent biofilm inhibition (Figure 18d) [143,144,145]. A set of thymidine derivatives bearing isoxazole and 1,2,3-triazole rings (TITL) were synthesized, and TITL-8f (Figure 18d) showed the best inhibitory activity against biofilm formation and virulence factors as a promising inhibitor [146]. A novel synthesized molecule, *N*-(2-pyrimidyl)butanamide (Figure 18d), was confirmed to reduce the expression of QS genes and was related to virulence factors in an animal embryo model [147]. A range of novel acetylene analogues of dihydropyrrolones from brominated dihydropyrrolones and acetylene analogues of fimbrolides brominated furanones were synthesized via Sonogashira coupling reactions. Biological test revealed that **6****2a**–**c**, **63a** and **6****3b** (Figure 18d) exhibited higher QS inhibitory activity against *P. aeruginosa* [4,148]. The flavone acacetin was investigated and showed strong inhibition capacity against the production of AHL-intermediated virulence factors and downregulated QS-related genes, but the exact action mechanism of it is still unknown [149]. Tenoxicam (Figure 18e) plays a role in the reduction of quorum-sensing-dependent antivirulence factor production, and therefore affects its pathogenesis in the host [150]. Diethylene triamine pentaacetic acid (DTPA, Figure 18e) is an FDA-approved drug that can also repress the production of elastase [151].

There are also many structurally uncertain substances with QS inhibitory abilities, and most substances are plant extracts. Extracts from four different Chinese herbal plants (*Poria cum Radix pini*, *Angelica dahurica*, *Rhizomacibotii*, and *Schizonepeta tenuifolia*) and polyphenolic extract from *Rosa rugosa* tea showede antibiofilm and quorum-sensing inhibitory potential [152,153]. The extracts of *Citrus sinensis*, *Laurus nobilis*, *Allium cepa*, *Elettaria cardamomum*, and *Coriandrum sativum* also show potent quorum-quenching effects on *P. aeruginosa* and eliminate pyocyanin formation [154]. In addition, *Acacia seyal* Del Bark extracts and plantain herb extracts affect QS-controlled extracellular virulence factors production [54,155]. Piper nigrum containing phenolic and flavonoid components, *Punica granatum* containing tannins and alkaloids, and *Pisum sativum* containing tannins and phenolic compounds have quorum-sensing inhibitory activity and can control the formation of biofilm [156]. Ferula oil and Dorema oil, belonging to the *Apiaceae* family, and grapefruit (*Citrus paradisi*) essential oils can produce flavonoids, which were found to reduce virulence factor production and interfere with the communication between bacteria [157,158]. Extracts from other natural sources such as the methanol extract of *T. bellerica* and extracts of *Oscillatoria subuliformis*, *Moringa oleifera*, *Cuphea carthagenensis*, *Coccinia indica*, and *Muntingia calabura* L. leaves, have been reported to interfere with the biofilm development and QS of *P. aeruginosa* [159,160,161,162,163,164]. The extracted metabolites of the endolichenic fungi (ELF) *Aspergillus quandricinctus* of *Usnea longissima* affect the activities of elastase and protease and prevent the biofilm formation of *P. aeruginosa* [165]. Many kinds of oils also can act as inhibitors. Rose, clove, and chamomile essential oils, lemon oil, and pine turpentine were found to work as QSIs [166,167]. Other compounds such as the aromadendrane-type sesquiterpenoid viridiflorol isolated from an Argentinian collection of the liverwort *Lepidoziachordulifera* showed potent biofilm formation inhibition against *P. aeruginosa* [167]. Polyphenols from *Salix tetrasperma* were reported to suppress virulence responses and inhibit the QS of *P. aeruginosa* [168].

## 7. QS Inhibitors from Marine Resources

The ocean is rich in resources, and researchers have reported that many compounds derived from marine organisms are biologically active. Marine sponges make up a large part of marine organisms, which makes them always the subject of research. Three secondary metabolites from marine sponges, manoalide, manoalide monoacetate, and secomanoalide (Figure 19a), were shown to be strong inhibitors of the *las* QS system [8]. Psammaplin A and bisaprasin (Figure 19b) are psammaplin-type compounds that derive from the marine sponge *Aplysinellarhax*; they can as QS inhibitors and show inhibitory effects on *las* and *rhl*, specifically on LasR and RhlR [169]. Marine plants are also good sources of QS inhibitors. The low molecular weight alginate oligomer (OligoG CF-5/20), which is produced by brown seaweed *Laminaria hyperborea*, can act as a QS antagonist to *las* and *rhl* QS pathway expression in *P. aeruginosa* and to inhibit biofilm formation [170]. Studies showed that an active constituent of seagrass, *Halodulepinifolia**,* 4-methoxybenzioic acid (Figure 19b), displayed inhibitory activity to QS-mediated virulence factors and biofilm formation [171]. Marine fungi also cannot be ignored as a main source. Equisetin (Figure 19c) was a tetra-mate-containing secondary metabolite of the marine fungus *Fusarium* sp. Z10 [18]. **6****4a**–**c** (Figure 19c) were isolated from the marine bacteria *Oceanobacillus* sp. XC22919 [172]. The active compounds tyramine and *N*-acetyltyramine were from marine bacteria [173]. These compounds were reported to show QS inhibition to *P. aeruginosa*. Therefore, compounds derived from marine organisms can be promising QS inhibitors and should be a mainstream subject in further research.

## 8. Conclusions

Quorum sensing is becoming a global concern; there have been many studies about QSIs. Among the existing reports, QS inhibitors targeting *las* account for the majority, and QS inhibitors targeting *rhl* are the least studied. There are many inhibitors that act on multiple systems. Many of the reported QSIs are from natural origins including plants and marine organisms. Apparently, marine biological resources are so rich that there are still many that we have not yet developed, and there are still many unrevealed QSIs from natural sources; therefore, it is worth continuing to explore more novel candidates. Another way to develop new QSIs is to modify the structures of natural products or to modify the structures of signal molecules, and long alkyl chains are important in transforming signal molecules. Moreover, designing compounds with the help of computational technology effectively provides ideas for the synthesis of new inhibitors. Meanwhile, it must be acknowledged that many effective inhibitors have unclear working models. Thus, studying the mechanisms of their bioactivities can be a future research direction. Some existing FDA-approved drugs were found to have inhibitory activity after restudying them. However, there are still many limitations to clinical use because there may be side effects. In general, it is meaningful to study QSIs, as it gives new hope for treating bacterial infections.

## Figures and Tables

**Figure 1 marinedrugs-20-00488-f001:**
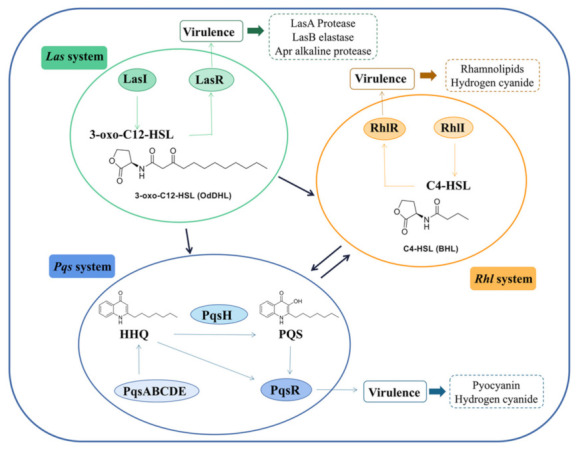
Three main QS systems in *P. aeruginosa*, *las* system, *rhl* system, and *pqs* system and their signal molecules. *Las* is indicated in green, *rhl* is indicated in orange, and *pqs* is indicated in blue. *Las* is at the top of the QS hierarchy and influences *rhl* and *pqs*. On the other hand, *rhl* is under the control of both *las* and *pqs*.

**Figure 2 marinedrugs-20-00488-f002:**
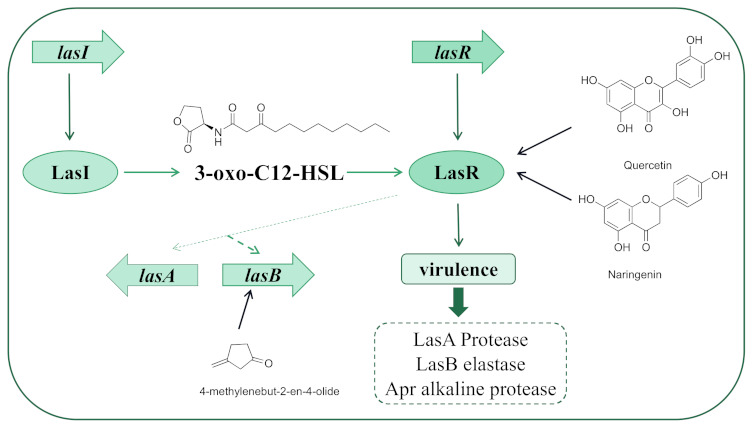
*Las* pathways with the QS controls of virulence and other phenotypes. Taking quercetin, naringenin, 4-methylenebut-2-en-4-olideas as examples, they have effects on LasR and *lasB*.

**Figure 3 marinedrugs-20-00488-f003:**
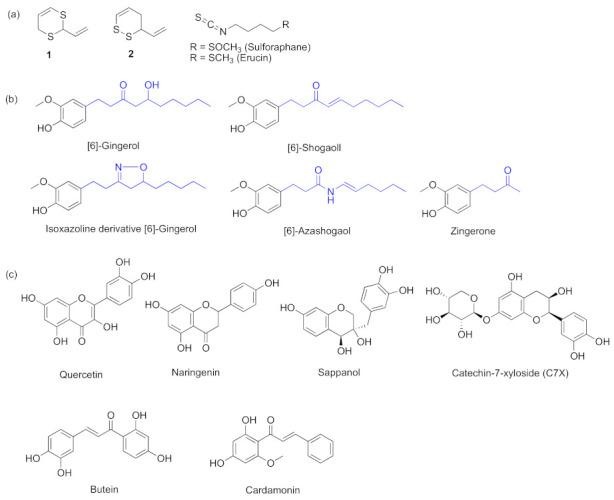
QS inhibitors from plants acting on *las*, including vinyl dithiins, isothiocyanates, phenolic compounds, and flavonoids. Groups highlighted in blue are important moieties related to QS inhibitory activity. (**a**) sulfur compounds; (**b**) key phenolic compounds of ginger; (**c**) flavonoids.

**Figure 4 marinedrugs-20-00488-f004:**
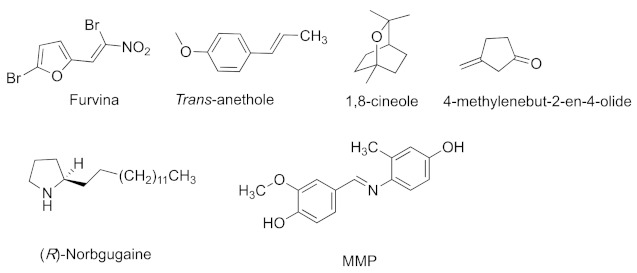
QS inhibitors from natural products acting on *las*.

**Figure 5 marinedrugs-20-00488-f005:**
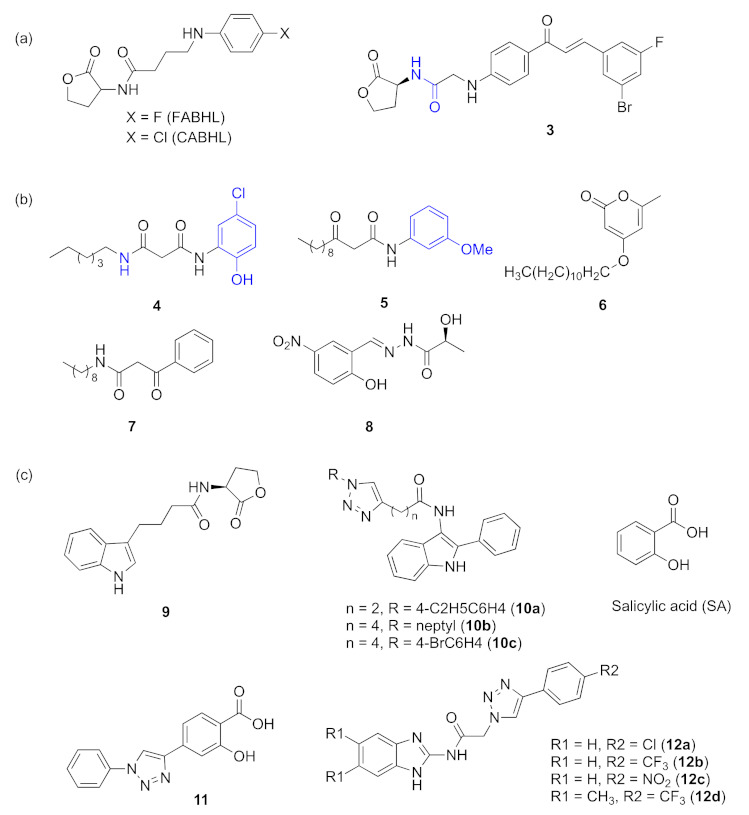
Synthetic compounds based on AHLs serve as QS inhibitors acting on *las*. Groups highlighted in blue are important moieties related to QS inhibitory activity. (**a**) structure of compounds similar to HSLs; (**b**) compounds containing a six membered ring; (**c**) indole based AHL analogues and salicylic acid analogues.

**Figure 6 marinedrugs-20-00488-f006:**
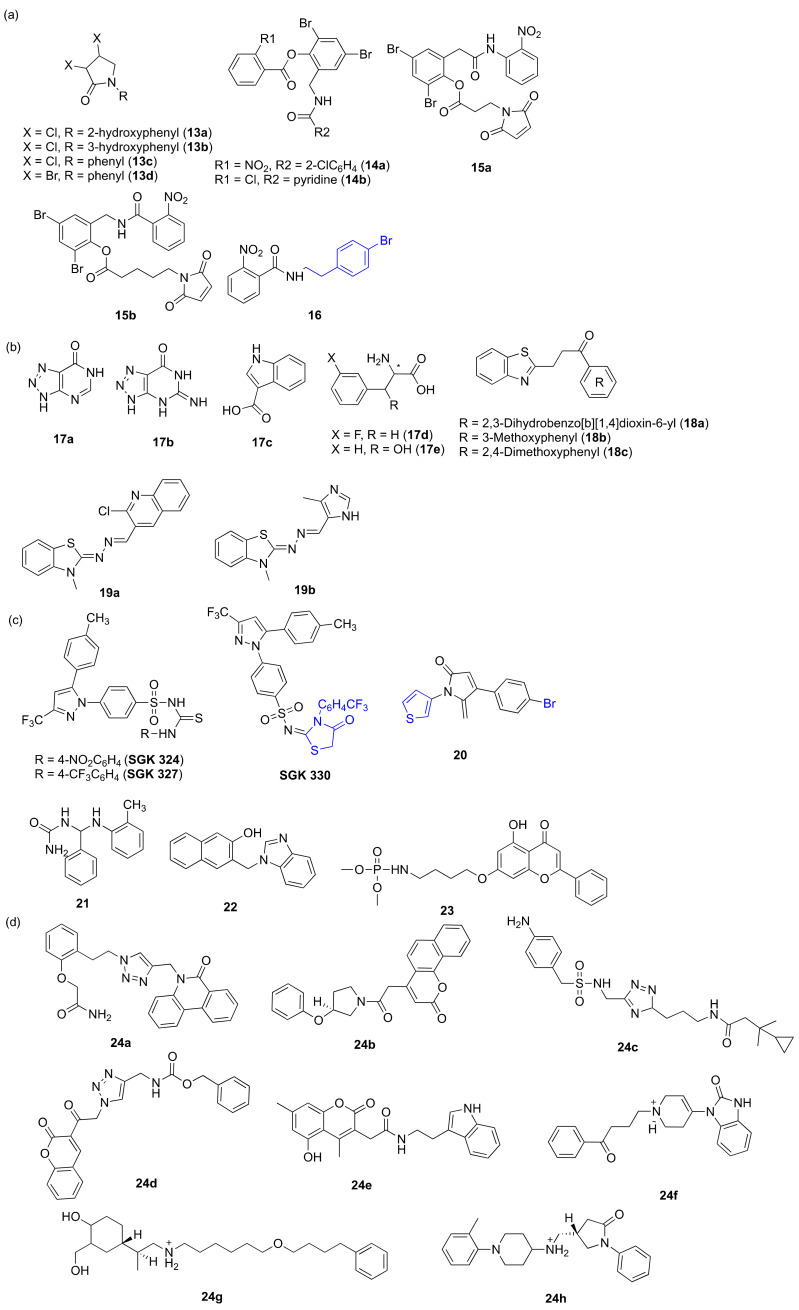
Inhibitors from synthetic compounds acting on *las*. Groups highlighted in blue are important moieties related to QS inhibitory activity. (**a**) compounds through reductive amination of mucochloric acid and mucobromic acid, compounds contain 2-nitrophenyl fragment; (**b**) computer-aided approaches designed compounds and compounds containing benzothiazole moiety; (**c**) unsymmetrical azines, Celecoxib derivatives, dihydropyrrolone analogues, Mannich bases and chrysin derivative; (**d**) compounds assessed employing a novel specialized multilevel in silico approach.

**Figure 7 marinedrugs-20-00488-f007:**
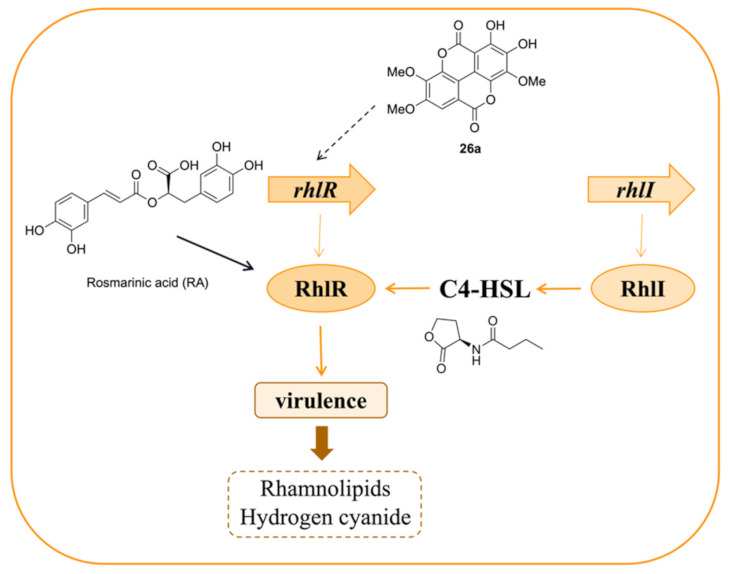
Rhl pathways with the QS controls of virulence and other phenotypes. Taking RA and **25a** as examples, they have effects on RhlR and *rhlR*, respectively.

**Figure 8 marinedrugs-20-00488-f008:**
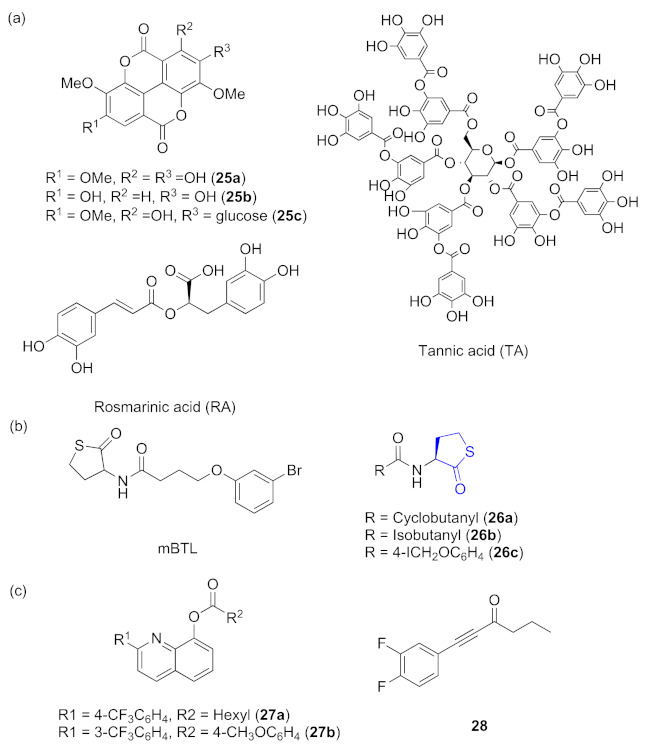
QS inhibitors acting on *rhl*. Groups highlighted in blue are important moieties related to QS inhibitory activity. (**a**) compounds from natural products; (**b**) AHL analogs; (**c**) synthetic compounds.

**Figure 9 marinedrugs-20-00488-f009:**
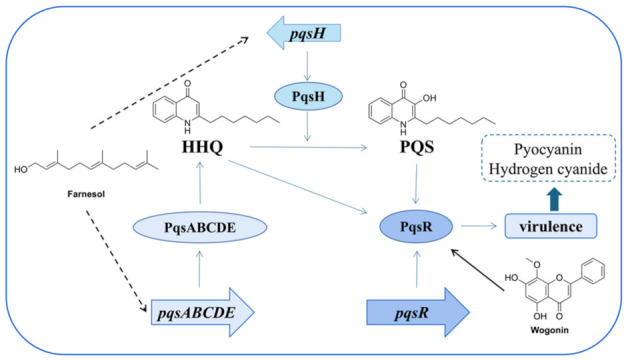
Pqs pathways with the QS controls of virulence and other phenotypes. Taking wogonin and farnesol as examples, they have effects on PqsR, *pqsH*, and *pqsABCDE*.

**Figure 10 marinedrugs-20-00488-f010:**
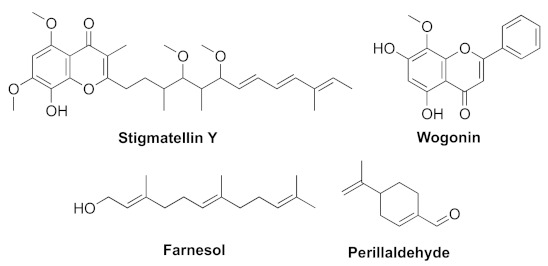
QS inhibitors from natural products acting on *pqs*.

**Figure 11 marinedrugs-20-00488-f011:**
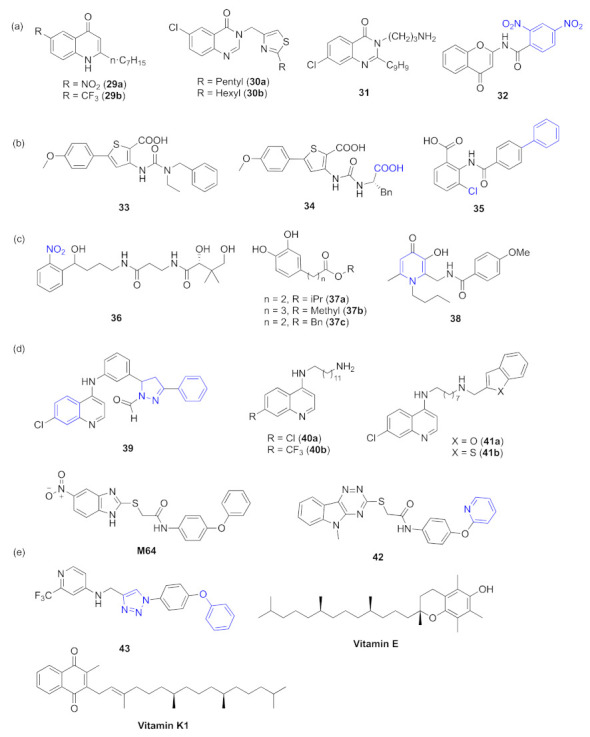
QS inhibitors from synthetic compounds acting on *pqs*. Groups highlighted in blue are important moieties related to QS inhibitory activity. (**a**) quinazolinones and chromones-based compounds; (**b**) compounds containing amide bonds or ureido motifs; (**c**) compounds containing a six membered ring; (**d**) compounds containing aminoquinolines and benzamide–benzimidazole series compounds. (**e**) miscellaneous.

**Figure 12 marinedrugs-20-00488-f012:**
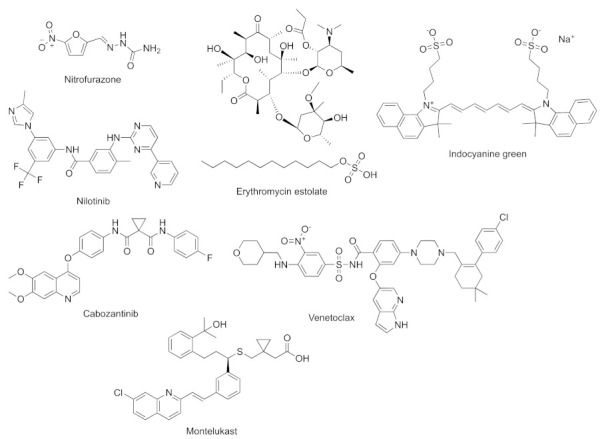
FDA-approved drugs that act as QS inhibitors.

**Figure 13 marinedrugs-20-00488-f013:**
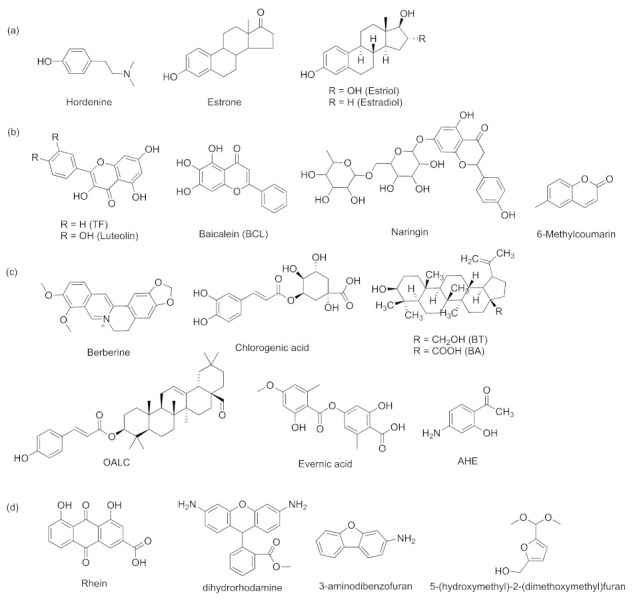
QS inhibitors acting on *las* and *rhl*. (**a**) human sex hormones and their structural relatives; (**b**) flavonoids; (**c**) alkaloids, triterpenoids, and phenolic compounds; (**d**) phytoconstituents from *Cassia fistula*.

**Figure 14 marinedrugs-20-00488-f014:**
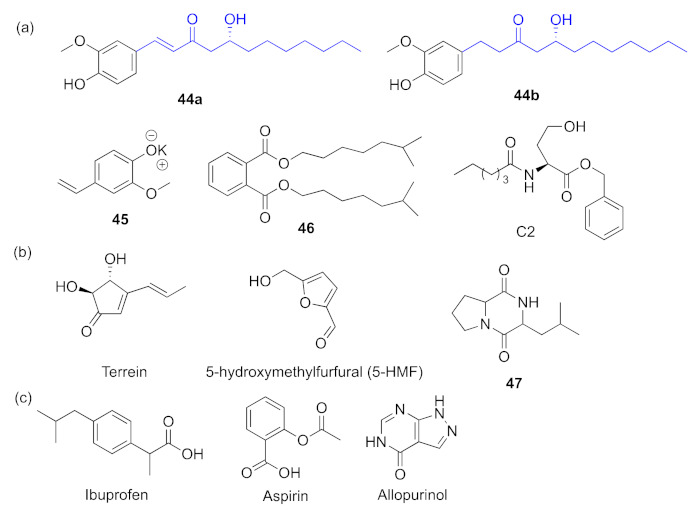
QS inhibitors acting on *las* and *rhl*. Groups highlighted in blue are important moieties related to QS inhibitory activity. (**a**) compounds containing a benzene ring; (**b**) compounds containing a five membered ring; (**c**) reported drugs.

**Figure 15 marinedrugs-20-00488-f015:**
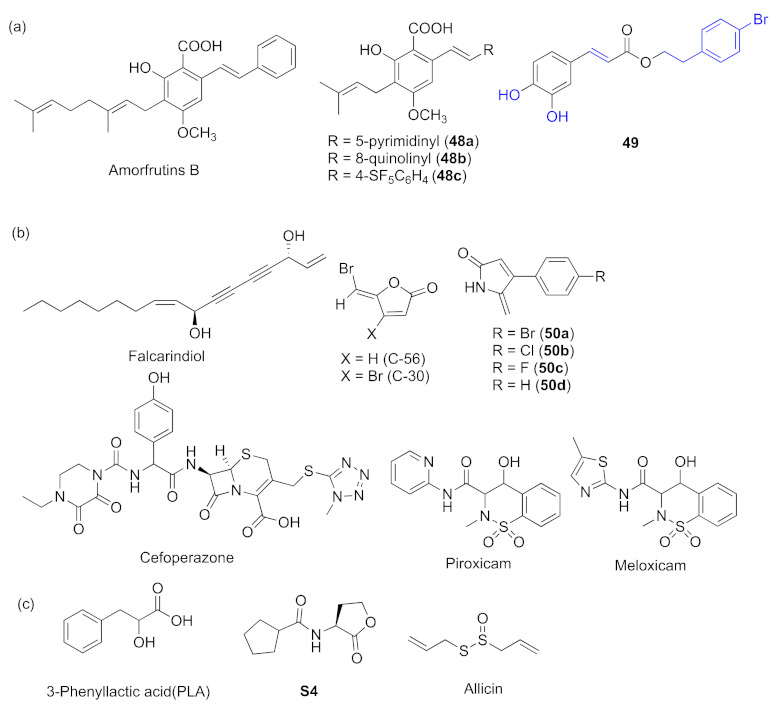
QS inhibitors acting on *las* and *pqs* or on *rhl* and *pqs*. Groups highlighted in blue are important related to showing QS inhibitory activity. (**a**) cajaninstilbene acid analogues and caffeic acid derivative (**b**) other compounds acting on *las* and *pqs*; (**c**) compounds acting on *rhl* and *pqs*.

**Figure 16 marinedrugs-20-00488-f016:**
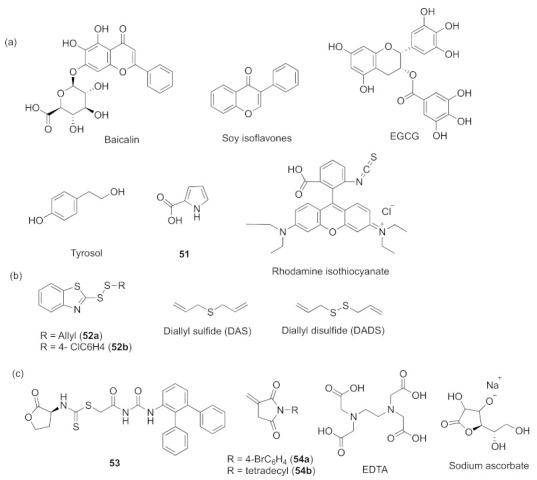
QS inhibitors acting on three systems. (**a**) compounds from natural sources; (**b**) sulfur-containing compounds; (**c**) miscellaneous.

**Figure 17 marinedrugs-20-00488-f017:**
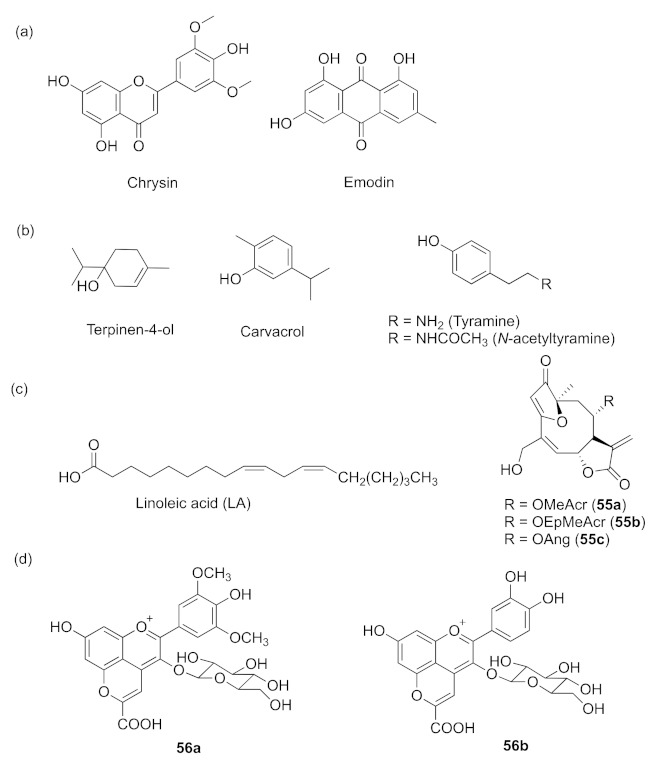
QS inhibitors from natural sources with undetermined working mechanisms. (**a**) flavone; (**b**) phenolic compounds and alcohols; (**c**) unsaturated polyenoids; (**d**) pyranoanthocyanins.

**Figure 18 marinedrugs-20-00488-f018:**
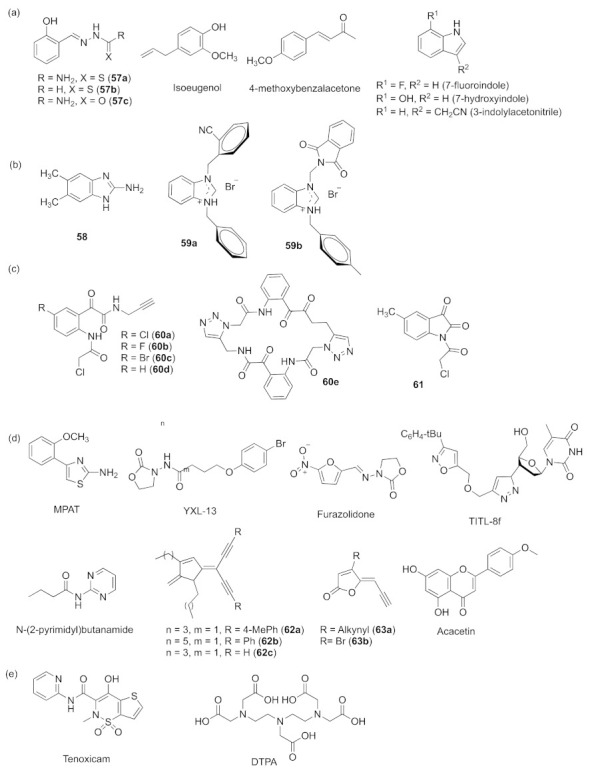
QS inhibitors through artificial synthesis with undetermined working mechanisms. (**a**) compounds with a benzene ring and indole derivatives; (**b**) benzimidazole derivatives; (**c**) glyoxamide derivatives; (**d**) miscellaneous; (**e**) Reported drugs.

**Figure 19 marinedrugs-20-00488-f019:**
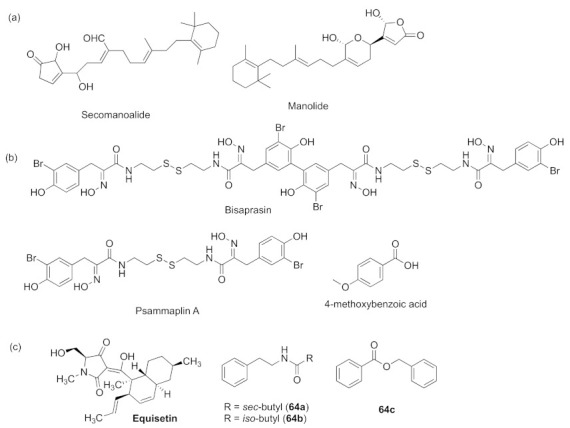
QS inhibitors from marine sources. (**a**) compounds from marine sponges; (**b**) compounds from marine sponge and seagrass; (**c**) compounds from marine microorganisms.

## Data Availability

Not applicable.

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
