# Peer review of "The Molecular Architecture of Pseudomonas aeruginosa Quorum-Sensing Inhibitors"

_marinedrugs, 2022, doi:10.3390/md20080488_

Round 1
Reviewer 1 Report
This manuscript is a comprehensive review manuscript on Pseudomonas aeruginosa quorum sensing (QS) inhibitors. While this review manuscript covered a lot of inhibitors reported to be involved with all three QS systems of the bacterium, which I mostly agree with, I feel that the QS systems of the bacteria itself are not adequately explained. Therefore, the followings are my suggestions that I think the authors should consider including in the manuscript.
The title should be changed to reflect the content; I suggest “Molecular Architecture of Pseudomonas aeruginosa Quorum Sensing Inhibitors.”
In the abstract: the first two sentences are hard to read. I suggest the authors rewrite them.
In the introduction, lines 27-28: “Quorum sensing (QS) system, also known as quorum quenching (QQ)” Are they the same?
In the introduction, lines 56-57: Among the three QS systems, las system is at the top of QS and is required for optimal activation of the rhl and pqs QS systems. What do the authors mean by “at the top of QS”?
Figure 1: The figure legend does not give enough detail. Please explain in detail how these three systems are related and what the different sizes and colors of arrows mean.
Figure 2. QS inhibitors from natural plants acting on las system, including vinyl dithiins, isothiocyanates, phenolic compounds and flavonoids.: The word “natural” should be removed.
In sections 2-4: Inhibitors of the three QS systems, I suggest the authors draw detailed pathways involved with the QS controls of virulence and other phenotypes with different compounds acting on (either inhibit or promote) these genes or proteins along the path. It will make these sections a lot easier to follow.
In section 3.4: “Inhibitors acting on rhl systems can be divided into three sources. Some inhibitors are from natural products.” What are the other two sources? Synthetic compounds and what?
Author Response
Referee: 1
Recommendation:
Comments:
This manuscript is a comprehensive review manuscript on Pseudomonas aeruginosa quorum sensing (QS) inhibitors. While this review manuscript covered a lot of inhibitors reported to be involved with all three QS systems of the bacterium, which I mostly agree with, I feel that the QS systems of the bacteria itself are not adequately explained. Therefore, the followings are my suggestions that I think the authors should consider including in the manuscript.
Q1:
The title should be changed to reflect the content; I suggest ‘Molecular Architecture of Pseudomonas aeruginosa Quorum Sensing Inhibitors’.
Reply: We have changed the title to ‘Molecular Architecture of Pseudomonas aeruginosa Quorum Sensing Inhibitors’.
Q2:
In the abstract: the first two sentences are hard to read. I suggest the authors rewrite them.
Reply: We have revised the first two sentences to ‘The survival selection pressure caused by antibiotic mediated bactericidal and bacteriostasis is one of the important inducements for bacteria to develop drug resistance. Bacteria gain drug resistance through spontaneous mutation so as to achieve the goals of survival and reproduction’.
Q3:
In the introduction, lines 27-28: "Quorum sensing (QS) system, also known as quorum quenching (QQ)" Are they the same?
Reply: They are different. Because the quorum quenching was not further explained in main text, we deleted the sentence ‘also known as quorum quenching (QQ)’.
Q4:
In the introduction, lines 56-57: Among the three QS systems, las system is at the top of QS and is required for optimal activation of the rhl and pqs QS systems. What do the authors mean by “at the top of QS”?
Reply: With las governing the expression of both pqs and rhl systems, it was often described as being at the top of the QS hierarchy.
Q5:
Figure 1: The figure legend does not give enough detail. Please explain in detail how these three systems are related and what the different sizes and colors of arrows mean.
Reply: We have added details and revised Figure 1. title to ‘Three main QS systems in P. aeruginosa, Las system, rhl system and pqs system and their signal molecules. Las system is indicated in green, rhl system is indicated in orange, pqs system is indicated in blue. Las system is at the top of the QS hierarchy, it has influences on rhl system and pqs system. On the other hand, rhl system is under the control of both las and pqs system’.
Q6:
Figure 2. QS inhibitors from natural plants acting on las system, including vinyl dithiins, isothiocyanates, phenolic compounds and flavonoids. The word “natural” should be removed.
Reply: In Figure 2. ‘QS inhibitors from natural plants acting on las system, including vinyl dithiins, isothiocyanates, phenolic compounds and flavonoids.’ The word ‘natural’ has been removed.
Q7:
In sections 2-4: Inhibitors of the three QS systems, I suggest the authors draw detailed pathways involved with the QS controls of virulence and other phenotypes with different compounds acting on (either inhibit or promote) these genes or proteins along the path. It will make these sections a lot easier to follow.
Reply: We have added figure 2 in section 2, figure 7 in section 3, figure 9 in section 4 to draw detailed pathways involved with the QS controls of virulence and other phenotypes with different compounds.
Q8:
In section 3.4: “Inhibitors acting on rhl systems can be divided into three sources. Some inhibitors are from natural products.” What are the other two sources? Synthetic compounds and what?
Reply: We have reorganized the sentence ‘Inhibitors acting on rhl systems can be divided into three sources: natural products, analogues of signaling molecules and synthetic compounds’.

Reviewer 2 Report
The manuscript provides a comprehensive overview on quorum sensing inhibitors against the pathogen Pseudomonas aeroginosa. Selection of literature for you is very good and comprises inhibitors against all three know quorum sensing pathways in P. aeroginosa. The authors missed the opportunity to present the data in a form allowing to easily draw chemical conclusion and help to get meta-level information from the digested literature. Therefore, I recommend to revise the manuscript in order to obtain this level of information. I propose to
- give information on methodological approaches to study the different pathways
- clearly indicate insights into the molecular achitecture of the different groups of inhibitors, deriving common molcular themes and identifying moeities and structures responsible for quroum sensing inhibition. This can be done by both, revising the graphs and inserting tables.
- quorum senising inhibition leads to different effects in the target bacterium (e.g. biofilm formation, etc.), a comprehensive comparison of the biological effects would be very helpful
- revise title and abstract to make clear that the manuscript focusses on P. aeroginosa - alternatively describe that the affected pathways are active in related microorganisms and that research results on P.a. can be transferred to other bacterial genera
- make clear, what the marine resources can contribute to the issue
Specifically I have the following comments:
- in the whole manuscript: all compound names in minor start letter, all species and genus names in italics
- in the whole mansucript: revise use of "could" ('Can' refers to a general truth or something that has a strong possibility. 'Could' refers to something that has a weak possibility, or something that might happen, but not necessarily a general truth.)
Title:
- specify P. aeroginosa a model organism
Abstract
- the abstract shall reflect the content and message of the full manuscript. please revise accordingly.
Body
- l 27 QS is not known as quorum quenching. This part of the sentence is wrong.
- l39 explain the D. pulchra findings (ie. the inhibitors are AHL analogues), as they give an idea on molecular structure underlying quorum sensing inhibition
-l47 delete "of plenty"
- l58ff the paragraph is not clear in its message and lacks references
- figure 1. explain abbreviations and interaction arrows; were is the reference to the figure in the text? also explain in text the interaction between the different systems: is the interaction based on compounds or proteins?
- l 84 or before: explain the methodological approach, how is specific las system inhibition measured and can be distinguished from the other quorum sensing pathways?- figure 2 (and all othe compound figures) indicate common structural elements, try to identify active moeities and pharmacophores graphically. For a number of molecules, the molecular docking points are described!- provide a table of the compounds, their origin, their target and the respective biological consequence/effect in P. aeroginosa- l214 delete "Whats more"- l222 explain the methodological approach, how is specifically rhl system inhibition measured and can be distinguished from the other quorum sensing pathways?- l235 list the three sources, it is hard to find in the subsequent text- l254 "shows"- l280 sentence seems incomplete after "anti-virulence"- l270 explain the methodological approach, how is specifically pqs system inhibition measured -l334 explain the computational analysis specifically done for pqs acting drugs. which target was choosen?
- l348 remove "opportunistic pathogen"- l370 "repress"instead of "depress"- l374 typo- l436ff elaborate on the mechanism- l531ff extracts are not substances! revise the sentence accordingly- l557 "marine sources"?- l557ff the compounds should be sorted into the respective chapter (las, rhs etc), if the authors whis specific information on the marine sources they should rather elaborate on the potential and specificitiesconclusion- The conclusion lacks the learning from the data - one can conclude on core structures, on phamacological learnings, on limitations, on new approaches, options for new marine drugs etc...this chapter shall be revised.- l589 what is meant with "refining"References- carefully revise the references for typos, spelling and format
Author Response
Referee: 2
Recommendation:
Comments:
The manuscript provides a comprehensive overview on quorum sensing inhibitors against the pathogen Pseudomonas aeroginosa. Selection of literature for you is very good and comprises inhibitors against all three know quorum sensing pathways in P. aeroginosa. The authors missed the opportunity to present the data in a form allowing to easily draw chemical conclusion and help to get meta-level information from the digested literature. Therefore, I recommend to revise the manuscript in order to obtain this level of information.
Q1:
in the whole manuscript: all compound names in minor start letter, all species and genus names in italics
Reply: We have revised all compound names in minor start, all species and genus names in italics.
Q2:
in the whole manuscript: revise use of "could" ('Can' refers to a general truth or something that has a strong possibility. 'Could' refers to something that has a weak possibility, or something that might happen, but not necessarily a general truth.)
Reply: In the whole manuscript, we have revised those ‘could’ to ‘can’, which indicating a general truth.
Q3:
Title: specify P. aeroginosa a model organism
Reply: We have changed the title to ‘Molecular Architecture of Pseudomonas aeruginosa Quorum Sensing Inhibitors’.
Q4:
the abstract shall reflect the content and message of the full manuscript. please revise accordingly.
Reply: We have revised the whole abstract as follows: The survival selection pressure caused by antibiotic mediated bactericidal and bacteriostatic is one of the important inducements for bacteria to develop drug resistance. Bacteria gain drug resistance through spontaneous mutation so as to achieve the goals of survival and reproduction. Quorum sensing (QS) system is an intercellular communication system based on cell density and can regulate bacterial virulence and biofilm formation. The secretion of more than 30 virulence factors of P. aeruginosa is controlled by QS, and the formation and diffusion of biofilm is an important mechanism causing multidrug resistance of P. aeruginosa, which is also closely related to QS system. There are three main QS systems in P. aeruginosa: las system, rhl system and pqs system. Quorum sensing inhibitors (QSIs) can reduce the toxicity of bacteria without affecting the growth and enhance the sensitivity of bacterial biofilms to antibiotic treatment. These characteristics make QSIs become a hot research and development in the field of anti-infection. This paper reviews the research progress of P. aeruginosa quorum-sensing system and QSIs targeting three QS systems, which will provide help for the future research and development of novel quorum-sensing inhibitors.
Q5:
l27 QS is not known as quorum quenching. This part of the sentence is wrong.
Reply: In line 29, we have revised the sentence to ‘Quorum sensing (QS) system is a well-known cell to cell signal communication system that allows bacteria to monitor their cell density by releasing signaling molecules called autoinducers (AIs) to cope with the changes in society and environment’.
Q6:
l39 explain the D. pulchra findings (ie. the inhibitors are AHL analogues), as they give an idea on molecular structure underlying quorum sensing inhibition
Reply: In line 40, we have explained the D. pulchra findings and revised to ‘The initial demonstration of quorum sensing inhibitors (QSIs) can be traced back to Australian red marine algae Delisea pulchra, it produces secondary metabolites, a number of halogenated furanones, which have structural similarities to AHL molecules, these metabolites can interfere with bacterial processes which involve AHL-driven quorum-sensing systems’.
Q7:
l47 delete "of plenty"
Reply: In line 49, we have deleted "of plenty".
Q8:
l58 the paragraph is not clear in its message and lacks references
Reply: We have added references to the paragraph in line 61.
Q9:
figure 1. explain abbreviations and interaction arrows; where is the reference to the figure in the text? also explain in text the interaction between the different systems; is the interaction based on compounds or proteins?
Reply: We have added reference [11] in the text; We have revised Figure 1. title to "Three main QS systems in P. aeruginosa, Las system, rhl system and pqs system and their signal molecules. Las system is indicated in green, rhl system is indicated in orange, pqs system is indicated in blue. Las system is at the top of the QS hierarchy, it has influences on rhl system and pqs system. On the other hand, rhl system is under the control of both las and pqs system"; The interaction is based on proteins and signal molecules.
Q10:
l84 or before: explain the methodological approach, how is specific las system inhibition measured and can be distinguished from the other quorum sensing pathways?
Reply: We have explained the methodological approach in section 2 ’Specific las system inhibition is measured through various approaches, for example, RNA extraction and microarray analysis, high-throughput RNA sequencing, molecular docking experiment, gene expression assay, quantitative real-time PCR and so on. Specific rhl and pqs system inhibition are measured through similar approaches with las system’.
Q11:
figure 2 (and all other compound figures) indicate common structural elements, try to identify active moieties and pharmacophores graphically. For a number of molecules, the molecular docking points are described!
Reply: We have revised the compounds in figures, and functional groups highlighted in blue are important moieties in showing QS inhibitory activity.
Q12:
provide a table of the compounds, their origin, their target and the respective biological consequence/effect in P. aeroginosa.
Reply: We divided our main sections according to targets and illustrated origins and the respective biological consequence/effect in P. aeruginosa. We tried to give a table but found the current figures overlap the information. Thus, we decided to keep the original sequence of the topic. Thanks for your suggestion and understanding.
Q13:
l214 delete "What's more"
Reply: In line 214, we have deleted "What's more".
Q14:
l222 explain the methodological approach, how is specifically rhl system inhibition measured and can be distinguished from the other quorum sensing pathways?
Reply: We have explained the methodological approach in section 2 :"Specific las system inhibition is measured through various approaches, for example, RNA extraction and microarray analysis, high-throughput RNA sequencing, molecular docking experiment, gene expression assay, quantitative real-time PCR and so on. Specific rhl and pqs system inhibition are measured through similar approaches with las system".
Q15:
l235 list the three sources, it is hard to find in the subsequent text
Reply: In line 244, we have revised the sentence ‘Inhibitors acting on rhl systems can be divided into three sources’ to ‘Inhibitors acting on rhl systems can be divided into three sources: natural products, analogues of signaling molecules and synthetic compounds’.
Q16:
l254 "shows"
Reply: In line 254, we have changed "show" to " shows".
Q17:
l280 sentence seems incomplete after ‘anti-virulence’
Reply: In line 301, we have revised the sentence ‘It has been found having ability to anti-virulence and repressing biofilm formation’ to ‘It has been found having ability to inhibit virulence factors and repressing biofilm formation’.
Q18:
l270 explain the methodological approach, how is specifically pqs system inhibition measured
Reply: We have explained the methodological approach in section 2 :"Specific las system inhibition is measured through various approaches, for example, RNA extraction and microarray analysis, high-throughput RNA sequencing, molecular docking experiment, gene expression assay, quantitative real-time PCR and so on. Specific rhl and pqs system inhibition are measured through similar approaches with las system".
Q19:
l334 explain the computational analysis specifically done for pqs acting drugs. which target was chosen?
Reply: Docking and virtual screening experiments were applied to discover new inhibitors targeting PqsR, molecular dynamic simulations and MM/GBSA free-energy calculations were performed to confirm the docking predictions and elucidate on the mode of interaction.
Q20:
l348 remove "opportunistic pathogen"
Reply: In line 348, we have removed "opportunistic pathogen".
Q21:
l370 "repress" instead of "depress"
Reply: In line 391, we have changed "depress" to "repress".
Q22:
l374 typo
Reply: In line 395, we have revised comma in English font.
Q23:
l436 elaborate on the mechanism
Reply: 3-Phenyllactic acid strongly binds to receptors RhlR and PqsR. Allicin inhibits C4-HSLs synthesis-related gene rhlI. It also suppresses the expression levels of genes like rhlA, rhlB, and rhlC, which is related to rhamnolipid synthesis and inhibits PQS molecule synthesis. We consulted the literature, the action mechanism of S4 was not elaborated clearly, the literature shows that stimulation of the rhl system by RhlR agonists S4 can strongly suppress pqs signaling in the wild-type organism and it was showed that it can disrupt rhl-pqs crosstalk.
Q24:
l531 extracts are not substances! revise the sentence accordingly
Reply: In line 510, we have revised the sentence "extracts Tyramine and N-acetyltyramine" to "active compounds Tyramine and N-acetyltyramine from marine bacteria".
Q25:
l557 "marine sources"?
Reply: In line 584, we have changed "marine sources" to "marine organisms".
Q26:
l557 the compounds should be sorted into the respective chapter (las, rhs etc), if the authors wish specific information on the marine sources they should rather elaborate on the potential and specificities conclusion
Reply: There have great potential to develop QS inhibitors from marine organisms, for example, marine sponges, marine microorganisms and so on are all potential sources of quorum sensing inhibitors. The separate listing of inhibitors from marine sources corresponds to the theme of the journal: Marine Drugs.
Q27-Q28:
The conclusion lacks the learning from the data; One can conclude on core structures, on phamacological learnings, on limitations, on new approaches, options for new marine drugs etc...this chapter shall be revised.
Reply: We have revised the conclusion as follows: Quorum sensing is becoming a global concern, currently, there have been many studies about QSIs. Among the existing reports, QS inhibitors targeting las system account for the majority, QS inhibitors targeting rhl system are the least. There are many inhibitors that act on multiple systems. A major part of the reported QSIs are from natural origins including plant, marine organisms and so on. Apparently, marine biological resources are so rich that there are still many that we have not yet developed and there are still many unrevealed QSIs from natural sources, so it is worth keeping explore more novel candidates. Another way to develop new QSIs is to modify the structure of natural products or to modify the structure of signal molecule, and long alkyl chain is important in transforming signal molecules. What’s more, designing compounds with the help of computational technology is a good way. It provides ideas for the synthesis of new inhibitors. Meanwhile, we have to notice that many inhibitors, whose working model is still unclear. Thus, studying the mechanism of their bioactivities can be the future research directions. Some existing FDA approved drugs were found to have inhibitory activity after restudying them. However, there are still many limitations if it is used in clinic. There may be other side effects. In general, it is meaningful to study QSIs, as it gives new hope for treating bacterial infections.
Q29:
l589 what is meant with "refining"
Reply: We have changed "refining" to "restudying them".
Q30:
References: carefully revise the references for typos, spelling and format
Reply: We have revised the references for typos, spelling and format.

Reviewer 3 Report
Manuscript reviews the research progress of Pseudomonas aeruginosa quorum-sensing system, which will provide help for the future research and development of novel quorum-sensing inhibitors, which can reduce the toxicity of bacteria without affecting the growth and enhance the sensitivity of bacterial biofilms to antibiotic treatment. The review is up-to-date and relevant today, describes the work of the last decade and can be accepted for publication after a slight revision:
There is no reference to figure 1 in the text
Аuthors are encouraged to make a separate list of abbreviations at the end of the article
Author Response
Referee: 2
Recommendation: slight revisions required
Comments:
Manuscript reviews the research progress of Pseudomonas aeruginosa quorum-sensing system, which will provide help for the future research and development of novel quorum-sensing inhibitors, which can reduce the toxicity of bacteria without affecting the growth and enhance the sensitivity of bacterial biofilms to antibiotic treatment. The review is up-to-date and relevant today, describes the work of the last decade and can be accepted for publication after a slight revision:
Q1:
There is no reference to figure 1 in the text
Reply: We have added reference [11] in the text.
Q2:
Authors are encouraged to make a separate list of abbreviations at the end of the article.
Reply: We have added a separate list of abbreviations at the end of the review.

Round 2
Reviewer 2 Report
The manuscript was revised and is much more easy to follow. I still have an number of spelling and typing andtypesetting remarks, which I indicated in the attached file.
Contentwise I am happy with the manuscript and therefore I suggest minor revisions .

Author Response
Responses to Reviewer 2 (second version):
All typos have been amended accordingly.
In reference, some journals' abbreviations are correct, such as J. Am. Chem. Soc., Angew. Chem. Int. Ed., Front. Cell. Infect. Microbiol., Int. J. Mol. Sci., Nat. Prod. Res. and so on.
Please see attachments: manuscript with highlighting and manuscript in clean.